# Serological Testing Reveals the Hidden COVID-19 Burden among Health Care Workers Experiencing a SARS-CoV-2 Nosocomial Outbreak

Yu Nakagama,[a,b] Yuko Komase,[c] Katherine Candray,[a,b] Sachie Nakagama,[a,b] Fumiaki Sano,[d] Tomoya Tsuchida,[e] Hiroyuki Kunishima,[f] Takumi Imai,[g] Ayumi Shintani,[g] Yuko Nitahara,[a,b] Natsuko Kaku,[a,b] Yasutoshi Kido[a,b]

aDepartment of Parasitology, Graduate School of Medicine, Osaka City University, Osaka, Japan
bResearch Center for Infectious Disease Sciences, Graduate School of Medicine, Osaka City University, Osaka, Japan
cDepartment of Respiratory Internal Medicine, St. Marianna University School of Medicine, Yokohama City Seibu Hospital, Yokohama, Japan
dDepartment of Hematology and Oncology, St. Marianna University School of Medicine, Yokohama City Seibu Hospital, Yokohama, Japan
eDivision of General Internal Medicine, St. Marianna University School of Medicine, Kawasaki, Japan
fDepartment of Infectious Diseases, St. Marianna University School of Medicine, Kawasaki, Japan
gDepartment of Medical Statistics, Graduate School of Medicine, Osaka City University, Osaka, Japan

**ABSTRACT** We describe the results of testing health care workers, from a tertiary care hospital in Japan that had experienced a coronavirus disease 2019 (COVID-19) outbreak during the first peak of the pandemic, for severe acute respiratory syndrome coronavirus 2 (SARS-CoV-2)-specific antibody seroconversion. Using two chemiluminescent immunoassays and a confirmatory surrogate virus neutralization test, serological testing revealed that a surprising 42% of overlooked COVID-19 diagnoses (27/64 cases) occurred when case detection relied solely on SARS-CoV-2 nucleic acid amplification testing (NAAT). Our results suggest that the NAAT-positive population is only the tip of the iceberg and the portion left undetected might potentially have led to silent transmissions and triggered the spread. A questionnaire-based risk assessment was further indicative of exposures to specific aerosol-generating procedures (i.e., noninvasive ventilation and airway suctioning) having mediated transmission and served as the origins of the outbreak. Our observations are supportive of a multitiered testing approach, including the use of serological diagnostics, in order to accomplish exhaustive case detection along the whole COVID-19 spectrum.

**IMPORTANCE** We describe the results of testing frontline health care workers, from a hospital in Japan that had experienced a COVID-19 outbreak, for SARS-CoV-2-specific antibodies. Antibody testing revealed that a surprising 42% of overlooked COVID-19 diagnoses occurred when case detection relied solely on PCR-based viral detection. COVID-19 clusters have been continuously striking the health care system around the globe. Our findings illustrate that such clusters are lined with hidden infections eluding detection with diagnostic PCR and that the cluster burden in total is more immense than actually recognized. The mainstays of diagnosing infectious diseases, including COVID-19, generally consist of two approaches, one aiming to detect molecular fragments of the invading pathogen and the other to measure immune responses of the host. Considering antibody testing as one trustworthy option to test our way through the pandemic can aid in the exhaustive case detection of COVID-19 patients with variable presentations.

**KEYWORDS** COVID-19, SARS-CoV-2, health care workers, serology

Address correspondence to Yasutoshi Kido, kido.yasutoshi@med.osaka-cu.ac.jp.

When the coronavirus disease 2019 (COVID-19) pandemic began in January 2020, Japan was no exception to the rest of the world, where access to diagnostic testing was limited. Shortages in testing resources during the first wave of the pandemic

in spring 2020 compromised timely case detection and forced health care workers (HCWs) to work in a deep diagnostic fog. The situation caused frontline health care facilities to suffer unexpected severe acute respiratory syndrome coronavirus 2 (SARS-CoV-2) exposures, followed by nosocomial outbreaks. However, even after a profound increase in molecular testing capacity and an apparent clearance of the fog, SARS-CoV-2 continued to sneak through the shield of symptom-driven screening strategies (1). Infections free of symptoms (i.e., presymptomatic or asymptomatic infections) and thus left untested were hypothesized to constitute a major burden and to contribute to transmission (2).

In support of this hypothesis, reports from later active screening studies revealed a significant majority of SARS-CoV-2 infections to manifest atypical nonrespiratory presentations or even at times to remain asymptomatic (3). Such minimally symptomatic individuals, never to be suspected of having COVID-19, lack the opportunity to undergo SARS-CoV-2 nucleic acid amplification testing (NAAT) and, together with those with false-negative NAAT results, continue to carry the risk of becoming a source of transmission. COVID-19, having unprecedentedly heterogeneous pathology, constitutes a spectrum of disease resembling an iceberg. Behind the most severe, devastating pneumonia patients lies the large majority of patients who are only mildly symptomatic or even remain asymptomatic (4). Thus, NAAT alone is prone to overlooking the hidden burden, and multitiered testing with the use of various diagnostic modalities should aid in exhaustive case detection along the whole spectrum.

The incidence and origin of paucisymptomatic or asymptomatic cases forming a significant portion of the total COVID-19 burden remain to be fully elucidated. In this study, 414 HCWs at a tertiary care hospital in Japan were tested for SARS-CoV-2-specific antibody seroconversion approximately 2 months after facing an outbreak during the first wave of the pandemic in April and May 2020. The now-revealed, overall perspective of the total COVID-19 burden highlights the shocking previous underestimation and presents an important lesson to be learned in minimizing nosocomial spread and further enhancing preparedness against future pandemics.

## RESULTS

**Antibody seroconversion reveals the true burden of the nosocomial outbreak, which is underestimated by symptom-driven NAAT screening.** Of the 414 eligible and consenting HCWs, 186 (45%) of 414 underwent NAAT screening for SARS-CoV-2 during the active emergence of the hospital cluster of infections during April and May 2020. At that time, the approach to screening of at-risk HCWs for COVID-19 was symptom driven; therefore, the participants who had never undergone NAAT were those less prioritized due to their lacking either typical manifestations or occupational exposures to aerosol-generating procedures performed on suspected/confirmed COVID-19 patients. Thirty-seven (20% of those tested by NAAT and 8.9% of the entire HCW cohort) tested positive for SARS-CoV-2.

Approximately 2 months after the nosocomial outbreak had subsided, sera were collected from the participants and tested under the orthogonal testing algorithm (Fig. 1). NAAT and serological testing results are summarized in Table 1. Combining the NAAT-confirmed and serologically confirmed diagnoses, the total number of COVID-19 cases and the overall prevalence rate summed to 64 and 15% (64/414 HCWs), respectively. Symptom-driven NAAT screening had overlooked 42% (27/64 cases) of the definitive COVID-19 diagnoses. Of the HCWs who were serologically diagnosed, 23 (85%) of 27 had received negative NAAT results and 4 (15%) of 27 had never been suspected of having COVID-19 and thus had not undergone NAAT screening.

**Clinical presentation, mode of diagnosis, and magnitude of serological responses among the COVID-19 HCW cohort.** Demographic data for the COVID-19 cases within the HCW cohort of the present study are presented in Table 2. The mean age was 35 ±12 years, and 11 (17%) of 64 HCWs were male. Only 4 (6.3%) of 64 HCWs had known high-risk comorbidities (hypertension and/or diabetes mellitus), and 4.7% (3 of 64 HCWs) reported chronic steroid use. The majority of symptomatic COVID-19

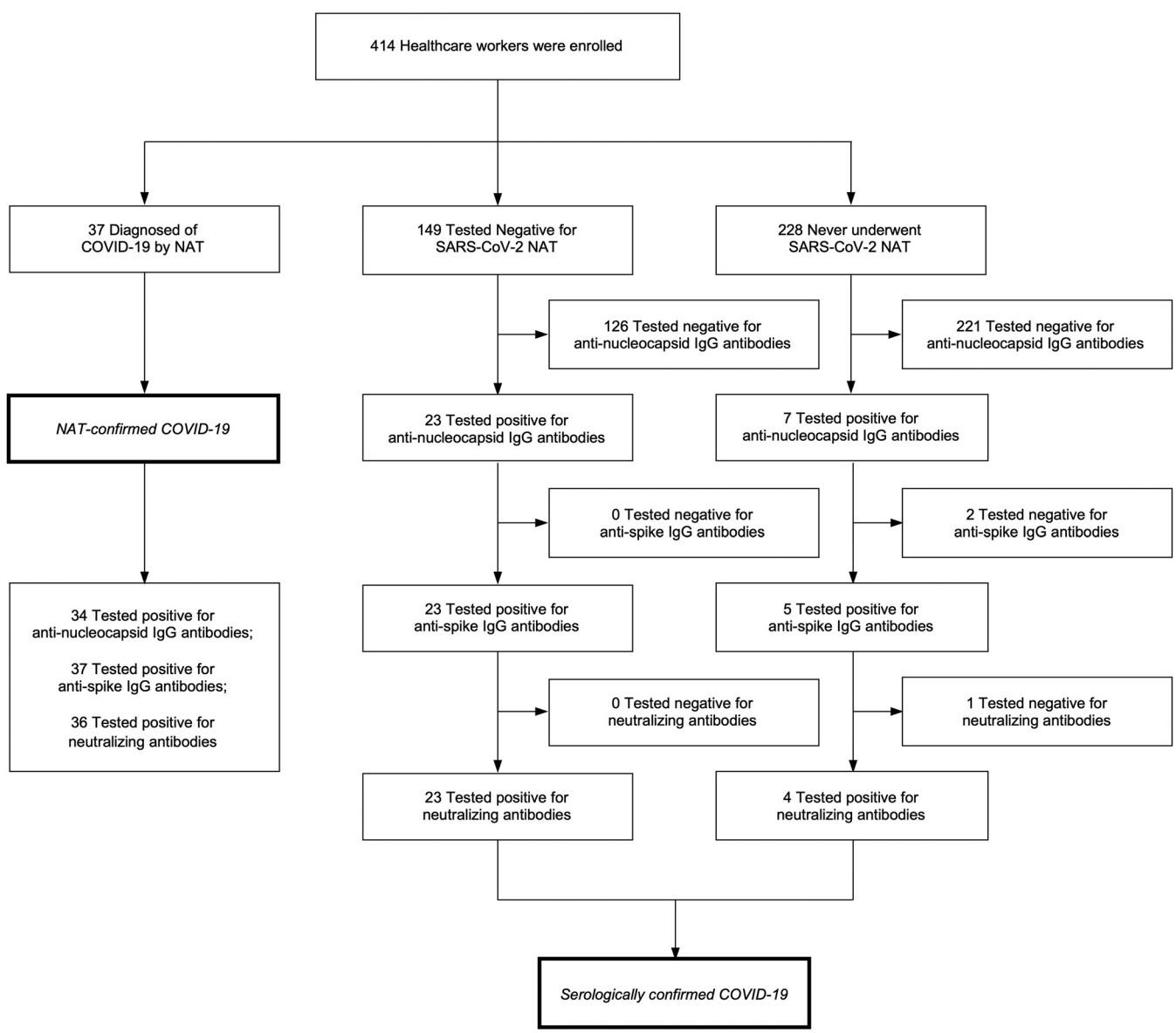

**FIG 1** Enrollment, results of testing, and algorithm for diagnosis. Of the 414 eligible and consenting participants, 186 had undergone NAAT for SARS-CoV-2. A total of 37 of the 186 tested HCWs were positive by NAAT. The orthogonal testing algorithm led to the detection of 27 excess COVID-19 cases that were diagnosed serologically. With NAAT- and serology-confirmed cases combined, the total number of COVID-19 diagnoses summed to 64.

cases were mild to moderate illnesses, and only 1 (1.6%) of 64 HCWs required $O_2$ supplementation, with no case fatalities. Typical respiratory symptoms were present in 31 (48%) of the 64 COVID-19 cases, and other cases presented with isolated hyposmia/anosmia (6 [9.4%] of 64 cases) or less specific systemic symptoms such as headache, abdominal symptoms, and/or malaise (8 [13%] of 64 cases). Notably, all 6 cases presenting with isolated hyposmia/anosmia were confirmed by NAAT (6 [100%] of 6

**TABLE 1** Summary of testing results for the subset of participants with available results for both NAAT and serological testing

| NAAT result | No. with serological test result of: | | Total no. |
|---|---|---|---|
| | Positive | Negative | |
| Positive | 33 | 4 | 37 |
| Negative | 23 | 126 | 149 |
| Total | 56 | 130 | 186 |

TABLE 2 Participant demographic data

| Demographic feature | COVID-19 cases (n = 64) | NAAT-confirmed cases (n = 37) | Serologically confirmed cases (n = 27) | P |
|---|---|---|---|---|
| Age (mean ± SD) (yr) | 35 ± 12 | 36 ± 12 | 33 ± 13 | 0.184 |
| Male (no. [%]) | 11 (17) | 7 (19) | 4 (15) | 0.748 |
| Preexisting risk condition (no. [%]) | | | | |
| Comorbidity | 4 (6.3) | 2 (5.4) | 2 (7.4) | 1.000 |
| Immunosuppressant use | 3 (4.7) | 2 (5.4) | 1 (3.7) | 1.000 |
| Severity | | | | |
| O$_2$ supplementation | 1 (1.6) | 1 (2.7) | 0 (0.0) | 1.000 |
| Death | 0 (0.0) | 0 (0.0) | 0 (0.0) | |
| Signs and symptoms (no. [%]) | | | | |
| Respiratory | 31 (48) | 26 (70) | 5 (19) | <0.001[a] |
| Hyposmia/anosmia | 6 (9.4) | 6 (16) | 0 (0.0) | 0.035[a] |
| Other | 8 (13) | 3 (8.1) | 5 (19) | 0.266 |
| None | 19 (30) | 2 (5.4) | 17 (63) | <0.001[a] |
| Imaging abnormality | 29 (45) | 21 (57) | 8 (30) | 0.079 |

[a]$P < 0.05$, $t$ test or Fisher's exact test.

cases). In contrast, asymptomatic cases (19 [30%] of 64 cases) were mainly confirmed by serological testing (17 [89%] of 19 cases).

Quantitative cross-comparison of the immune responses elicited (Fig. 2A) showed that the magnitude of immune responses targeting the two major nucleocapsid and spike antigens showed significant correlation within an individual (Spearman's $r = 0.67$; $P < 0.0001$). Further, compared with the levels of antinucleocapsid antibody titer (Spearman's $r = 0.56$; $P < 0.0001$), a stronger correlation between antispike antibody titers and surrogate virus

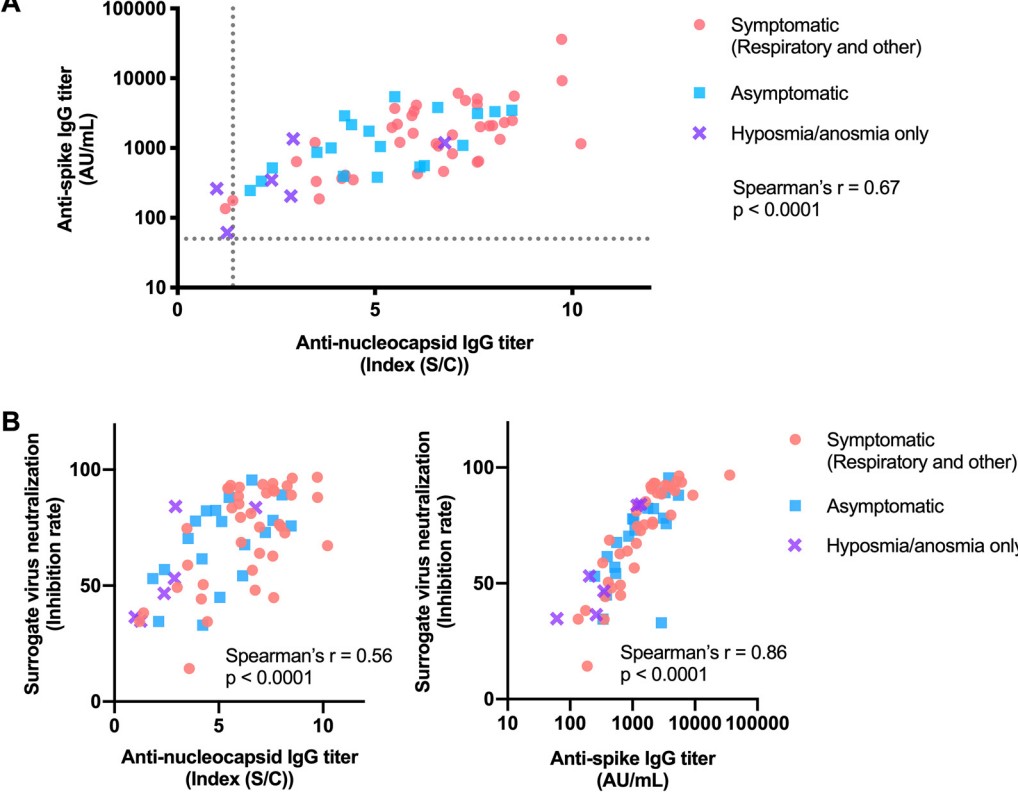

FIG 2 Quantitative assessment of serological responses and their mutual relationships. (A) Magnitudes of serological responses against the two major SARS-CoV-2 antigens. Dotted lines indicate the cutoff values. (B) In comparison with the antinucleocapsid IgG titer, the level of SARS-CoV-2 neutralizability, as assessed by the sVNT, was correlated with the antispike IgG titer to a greater extent.

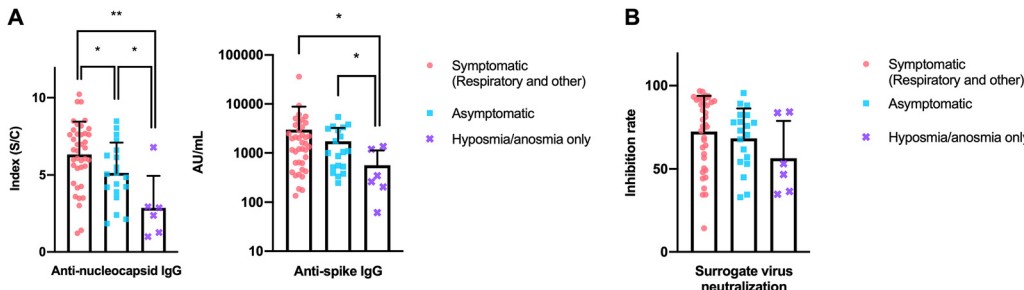

**FIG 3** Serological status of SARS-CoV-2 affected HCWs by symptom category. (A) HCWs with COVID-19 diagnoses who manifested isolated hyposmia/anosmia were characterized by diminished serological responses against the two major SARS-CoV-2 antigens. (B) A similar trend toward lower SARS-CoV-2 neutralizability of sera obtained from the 6 participants with isolated hyposmia/anosmia did not reach statistical significance. *, $P < 0.05$; **, $P < 0.01$, Mann-Whitney test. S/C, sample/cutoff.

neutralizability was observed (Spearman's $r = 0.86$; $P < 0.0001$) (Fig. 2B). Interestingly, compared with the other symptom categories, participants presenting with isolated hyposmia/anosmia elicited antinucleocapsid (symptomatic [respiratory and other] versus hyposmia/anosmia only, $P = 0.002$) and antispike (symptomatic [respiratory and other] versus hyposmia/anosmia only, $P = 0.014$) antibody responses of significantly lower magnitude, constituting an immunologically distinct subpopulation (Fig. 3A). Similarly, competition enzyme-linked immunosorbent assay (ELISA)-based surrogate virus neutralization tests (sVNTs) showed a trend toward lower neutralizability of the hyposmia/anosmia only subpopulation, although it did not reach statistical significance (symptomatic [respiratory and other] versus hyposmia/anosmia only, $P = 0.099$), possibly due to the small subset size ($n = 6$) (Fig. 3B).

**Defining procedural exposure-related risks.** Of the 414 eligible participants, 212 (51%) reported that they had participated in aerosol-generating procedures and thus had experienced SARS-CoV-2 exposures (Table 3). Among the variable types of aerosol-generating procedures, noninvasive ventilation (NIV) (relative risk [RR], 3.10 [$P = 0.008$]) conveyed the highest risk of SARS-CoV-2 transmission to the exposed HCWs, followed by airway suctioning (RR, 1.67 [$P = 0.040$]). Although sputum induction and cardiopulmonary resuscitation also seemed to convey substantial transmission risks to the exposed HCWs, the present study was underpowered to observe statistical significance for the risk increase related to these exposures. Although the procedural risk inherent to airway suctioning seemed substantially lower than that of NIV, airway suctioning, being a commonly performed aerosol-generating procedure, was the exposure to which the greatest number of excess COVID-19 cases were attributed (attributable number of events [AN], 16.0).

**TABLE 3** Risk of SARS-CoV-2 transmissibility during exposure to aerosol-generating procedures

| Procedural exposure status | No. (% of total) | No. (%) of COVID-19 cases | RR | RD | AFe | AN | P |
|---|---|---|---|---|---|---|---|
| Not exposed | 202 (49) | 24 (12) | Reference | Reference | | | |
| Exposed | 212 (51) | 40 (19) | 1.59 | 0.07 | 0.37 | 14.8 | 0.057 |
| Type of exposure | | | | | | | |
| Airway suctioning | 202 (49) | 40 (20) | 1.67 | 0.08 | 0.40 | 16.0 | 0.040[a] |
| NIV | 19 (4.6) | 7 (37) | 3.10 | 0.25 | 0.68 | 4.7 | 0.008[a] |
| Bag mask ventilation | 13 (3.1) | 0 (0.0) | | | | | 0.370 |
| Nebulizer administration | 8 (1.9) | 1 (13) | 1.05 | 0.01 | 0.05 | 0.05 | 1.000 |
| Sputum induction | 12 (2.9) | 4 (33) | 2.81 | 0.21 | 0.64 | 2.6 | 0.055 |
| O$_2$ supplementation via tracheostomy | 63 (15) | 8 (13) | 1.07 | 0.01 | 0.06 | 0.5 | 0.828 |
| Endotracheal intubation/extubation | 21 (5.1) | 2 (9.5) | 0.80 | −0.02 | −0.25 | −0.5 | 1.000 |
| Tracheostomy | 3 (0.7) | 0 (0.0) | | | | | 1.000 |
| Bronchoscopy | 0 (0.0) | 0 | | | | | |
| Cardiopulmonary resuscitation | 13 (3.1) | 3 (23) | 1.94 | 0.11 | 0.49 | 1.5 | 0.214 |

[a]$P < 0.05$, Fisher's exact test.

## DISCUSSION

The composite approach of combining NAAT- and serology-based diagnoses exhaustively detected definitive COVID-19 cases in the Japanese HCW cohort experiencing a nosocomial outbreak in April and May 2020. A surprising 42% of overlooked COVID-19 diagnoses occurred when case detection relied solely on NAAT, leading to undetected transmission. Taking note that the NAAT-positive population is only the tip of the iceberg and that a significant proportion of contagious individuals remain undetected, better allocation of testing resources is needed in order to clarify the true burden of COVID-19 and to reduce transmission in nosocomial settings.

NAAT-based case detection in Japan was counted on as a promising strategy, capable of thoroughly tracking SARS-CoV-2 transmissions and identifying and sizing infection clusters (1). It was not until June 2020, when the first national seroprevalence survey was performed, that the Japanese realized their 3- to 8-fold underestimation of the actual spread of the disease within the society (3). With the aim of enhancing case detection for effective quarantine, especially among presymptomatic or asymptomatic affected individuals, testing recommendations since then have shifted from a symptom-driven approach toward a mass-scale approach, which targets entire populations irrespective of symptoms. Against expectations, however, since it was the sole first-tier diagnostic method for this emerging infection, it is now increasingly recognized that NAAT-based SARS-CoV-2 pathogen detection has serious limitations. Because COVID-19 is primarily a lower respiratory tract disease, the probability of pathogen detection in upper respiratory tract specimens decreases rapidly and nearly halves within approximately 2 weeks after onset (5). Previous reports suggested that a substantial fraction (as much as up to 54%) of COVID-19 patients may present with undetectable viral loads and show false-negative reverse transcription (RT)-PCR results (6–8). Missed diagnoses having occurred not only in the paucisymptomatic and asymptomatic populations but also among acutely ill cases of high suspicion, indefinite molecular testing results already have left behind a significant burden of those in need of a diagnosis. A well-defined diagnostic method complementary to NAAT is still a serious need.

Since the host immune response lags behind viral invasion, the ability of antibody tests to detect an acute infection in its early phase is usually limited and is considered inferior to NAAT. In the case of COVID-19, however, NAAT performance itself remains suboptimal, and thus serological testing may aid in COVID-19 case detection by later identifying the infection during its subacute or chronic phase (9, 10). Serological tests show increasing diagnostic sensitivity with increasing time after the onset of symptoms. COVID-19 pneumonia with repeated false-negative NAAT results has been recurrently noted. In such clinical scenarios, serological testing with an extended detectable window has the potential to work complementarily with NAAT and to establish the diagnosis during the subacute phase of illness (9). Used in combination with NAAT, serological testing may enhance case detection and facilitate understanding of the actual spread of SARS-CoV-2 when applied to carefully targeted, high-risk populations, such as in-hospital outbreaks resembling the HCW cohort in the present study (10). By the use of well-designed immunoassays, as applied in the present study, serological tests may demonstrate, by day 21, an estimated overall sensitivity as high as 91.3% (95% confidence interval [CI], 82.3 to 95.9%) (11, 12). In addition, COVID-19-related long-lasting sequelae, such as anosmia or the multisystem inflammatory syndrome in children, are widely accepted as suitable indications for serological testing (13, 14).

In addition, exhaustive case detection has here enabled precision of risk estimates innate to aerosol-generating procedures. Our observations are in support of the prevailing concerns on the risks that aerosol-generating NIV may create for HCWs and provide implications regarding the origin of nosocomial spreads (15). The comparison of RR between procedures have demonstrated that the risk inherent to airway suctioning seemed substantially lower than NIV. However, the highest AN was associated with airway suctioning, indicating that this commonly performed aerosol-generating procedure could have contributed most significantly to scaling up the impact of the

outbreak. The above findings warn frontline HCWs about the harms of undervaluing risks related to any specific procedural exposure and stress once again the importance of being equipped with appropriate protectives when confronting novel pathogens.

Cross-comparison of the participants' serological responses highlighted the heterogeneity in width and magnitude of humoral immune responses among the affected. The observed higher detection rate among isolated hyposmia/anosmia patients by NAAT and the uniquely suppressed humoral immune response of the subpopulation may be reflecting confined viral replication and subsequent localized host immune reactions in the nasal airway. However, to draw conclusions regarding the relationships between viral tropism and serological responses of the host, data laying emphasis on individuals presenting with isolated hyposmia/anosmia are still lacking. Therefore, it remains a future consideration to refine pretest probabilities and to individualize diagnostic approaches based on case presentation (16, 17).

Limitations of the present study are mentioned below. First, since the interpretation of the serological status of the participants was not based on comparison with a paired serum sample drawn prior to the outbreak, we could not fully rule out the possibility that some of the participants had acquired their SARS-CoV-2 infections within the community, before the occurrence of the nosocomial outbreak. Similarly, a 2-month lag from the outbreak preceded the serum collection for serological testing, during which some of the participants might have acquired their SARS-CoV-2 infections outside the hospital setting. However, given that the SARS-CoV-2 seroprevalence in Japan was as low as 0.1% until June 2020, the likelihood of any of the participants having become infected in either of the aforementioned situations, independent of the outbreak in April and May 2020, was estimated to be very low. In addition, the health status of the HCWs was kept under continuous and intense monitoring, so that any potential symptomatic SARS-CoV-2 infection that could have occurred outside the hospital while awaiting serum collection would have been successfully identified, tested, and reported to the researchers. Taken together, these conditions led us to reasonably assume that the great majority of the seropositive individuals participating in the study had acquired their infections in the in-hospital setting during the outbreak. Second, since the indication for NAAT was prioritized among participants based on the presence of symptoms and/or close contacts, tests were never performed (and thus their results were not available) for 228 of the participants. Therefore, there remains a possibility that NAAT, had it been performed in a more scaled manner, could have led to the detection of additional COVID-19 cases. This could potentially have caused an underestimation of the clinical sensitivity of our NAAT assay protocol to efficiently detect COVID-19 cases. Finally, 4 of the 37 NAAT-positive individuals had negative serological test results, indicating discrepancies between NAAT and serological test results. Although SARS-CoV-2 antibody titers of the 4 individuals were above the cutoff value according to two of the three serological assays, they all fell just short of fulfilling our strictly set criteria for a serologically confirmed COVID-19 diagnosis, since the result of the third test was not supportive of a positive result. The ~2-month interval before serum sample collection might have led to a decline in their antibody titers and affected the interpretation of their serological status.

In conclusion, by way of analyzing serological status for SARS-CoV-2, we detected the missed diagnoses among HCWs from a tertiary care hospital in Japan that had experienced a COVID-19 outbreak during the first peak of the pandemic. Our observations here emphasize the efficiency of well-designed serological diagnostics in the detection of COVID-19 cases and SARS-COV-2 transmissions and indicate that the true spread within the hospital was even more extensive than previously estimated using symptom-based NAAT surveillance. Multitiered diagnostics are key to tracing the exact COVID-19 burden and, without consideration of the hidden but significant portion of the iceberg beneath the surface, we face the risks of underestimating COVID-19 disease prevalence, overestimating death rates, and misinterpreting exposure-specific risks.

## MATERIALS AND METHODS

**Cohort and samples.** A total 414 HCWs at St. Marianna University School of Medicine, Yokohama City Seibu Hospital (Kanagawa, Japan) who gave consent to participating in the study were recruited. Sera were obtained from the entire cohort within 3 consecutive days, from 30 June to 2 July 2020, when approximately 2 months had passed since the nosocomial outbreak in April and May 2020. Among the individuals with a known date of COVID-19 diagnosis, the interval between the date of diagnosis and the date of serum sampling ranged from 6 to 10 weeks. Analyses were conducted in accordance with the ethical standards noted in the 1964 Declaration of Helsinki and its later amendments. The research was approved by the Osaka City University Institutional Ethics Committee (approval 2020-003). Consent for participation and publication was obtained from every participant.

**Molecular testing.** NAAT for SARS-CoV-2 detection was performed using nasal swabs, based on the RT-PCR protocol developed by the National Institute of Infectious Diseases, Japan. The method targets two sites of the nucleocapsid gene (18), with reported sensitivity and specificity of 100% (95% CI, 94.7 to 100%) and 100% (95% CI, 95.8 to 100%), respectively (19).

**Serological testing.** Two chemiluminescent immunoassays, the SARS-CoV-2 IgG and SARS-CoV-2 IgG II Quant assays (Abbott, IL, USA), designed to detect serum IgG antibodies targeting the nucleocapsid and spike proteins, respectively, of SARS-CoV-2 were performed in accordance with the manufacturer's instructions; signals equal to or above cutoff values of 1.4 (index [sample/control]) and 50 AU/ml, respectively, were considered serologically positive. An orthogonal testing algorithm was adopted in order to optimize positive predictivity and to determine, with high specificity, the individuals who were truly seropositive for SARS-CoV-2-specific antibodies (3). In a previous study, the orthogonal approach was adopted to determine the seroprevalence for SARS-CoV-2 in Japan in June 2020. Although they were not mentioned in the original article, the sensitivity and specificity and their 95% CIs (Wilson/Brown method) for the algorithm were recalculated using available data. The sensitivity and specificity for the algorithm recalculated using available data were 100% (95% CI, 68 to 100%) and 100% (95% CI, 94 to 100%), respectively. In the algorithm, the individuals who initially tested positive for antinucleocapsid antibodies were tested with a second test targeting the SARS-CoV-2 spike antigen. For participants who were positive for both SARS-CoV-2-specific antibodies, a serological diagnosis of COVID-19 was finally confirmed by detecting neutralizing antibodies against SARS-CoV-2 using the SARS-CoV-2 sVNT (GenScript, Leiden, Netherlands), a competition ELISA-based sVNT. An inhibition rate of ≥30%, which, according to the manufacturer's instructions, is predictive of a half-maximal plaque reduction neutralization titer of ≥20, was selected as the cutoff value to determine positivity for neutralizing antibodies.

**COVID-19 case definitions.** Participants were defined as definitive COVID-19 patients when they were either (i) positive by NAAT (NAAT-confirmed COVID-19) or (ii) confirmed as serologically positive by the orthogonal testing algorithm (serologically confirmed COVID-19).

**Questionnaire for procedural exposure risk assessment.** Participants completed a questionnaire that included demographic data, medical history, occupational exposure to aerosol-generating procedures performed on patients with confirmed COVID-19, presence/absence of symptoms compatible with COVID-19, and state of NAAT diagnosis. The procedural exposures of interest in this study were participation in (i) airway suctioning, (ii) NIV, (iii) bag mask ventilation, (iv) nebulizer administration, (v) sputum induction, (vi) oxygen supplementation as part of tracheostomy care, (vii) endotracheal intubation or extubation, (viii) tracheostomy, (ix) bronchoscopy, and (x) cardiopulmonary resuscitation.

**Statistical analysis.** The results of molecular or serological testing were described as frequencies and percentages among the participants screened. To assess the differences among demographic characteristics between NAAT-confirmed and serologically confirmed COVID-19 patients, the following demographic variables were compared by $t$ tests (for the age variable) or Fisher's exact test (for the other variables, including male sex, preexisting risk condition, severity, and signs and symptoms). The magnitudes of serological responses against the nucleocapsid and spike antigens and the inhibition rate in the sVNT assay were compared by the Mann-Whitney test according to symptom category, i.e., participants with respiratory and/or systemic symptoms (symptomatic), participants expressing no symptoms (asymptomatic), and participants complaining of isolated smell impairments (hyposmia/anosmia only). Spearman's correlation coefficient was calculated for the various indices of serological responses. For the procedural exposure risk assessment, the risk ratio (RR) and risk difference (RD) for each exposure were calculated as the ratio or the absolute difference, respectively, between the COVID-19 incidence among those exposed to the aerosol-generating procedures and that in the reference (not exposed) group. The association between exposures to aerosol-generating procedures and COVID-19 incidence was tested by Fisher's exact test. To evaluate the extent of harm attributable to each procedure regarding the actual increase in COVID-19 cases, the attributable fraction among the exposed (AFe) and the AN were calculated. AFe is the proportion of COVID-19 diagnoses in the exposed group attributable to the occupational exposure and was calculated for each exposure as AFe = (RR − 1)/RR (20). AN is the absolute number of COVID-19 diagnoses attributable to the occupational exposure and was calculated for each exposure as AN = AFe × (number of COVID-19 diagnoses among the exposed). $P$ values of <0.05 were considered statistically significant.

## ACKNOWLEDGMENTS

This research was supported by the Japan Agency for Medical Research and Development (AMED) under grants JP20wm0125003 (Yasutoshi Kido), JP20he1122001 (Yasutoshi Kido), JP20nk0101627 (Yasutoshi Kido), and JP20jk0110021 (Yu Nakagama).

We receive financial support from the Special Reserves Fund for COVID-19 (Osaka City University) and the COVID-19 Private Fund (Shinya Yamanaka Laboratory, Center for iPS Cell Research and Application, Kyoto University). Yuko Nitahara is a recipient of the BIKEN Taniguchi Scholarship.

Minako Hosokawa, Hiroko Tanaka, Tomoyo Tominaga, and Harumi Domyo from the St. Marianna University School of Medicine, Yokohama City Seibu Hospital, supported the questionnaire distribution and sample and data collection. Reagents for serological testing were provided by Abbott Japan, LLC.

Yasutoshi Kido and Yu Nakagama report receiving financial support from Abbott Japan, LLC.

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
