## [Reviewer comments · Microbiology Spectrum]

**Microbiology
Spectrum**

Serological testing reveals the hidden COVID-19 burden among healthcare workers experiencing a SARS-CoV-2 nosocomial outbreak.

Yu Nakagama, Yuko Komase, Katherine Candray, Sachie Nakagama, Fumiaki Sano, Tomoya Tsuchida, Hiroyuki Kunishima, Takumi Imai, Ayumi Shintani, Yuko Nitahara, Natsuko Kaku, and Yasutoshi Kido

Corresponding Author(s): Yasutoshi Kido, Osaka City University

Review Timeline:

Submission Date:	July 28, 2021
Editorial Decision:	August 4, 2021
Revision Received:	August 17, 2021
Accepted:	August 18, 2021

Editor: S. Wesley Long

Reviewer(s): The reviewers have opted to remain anonymous.

Transaction Report:

DOI: <https://doi.org/10.1128/Spectrum.01082-21>

August 4, 2021

Dr. Yasutoshi Kido
Osaka City University
Osaka
Japan

Re: Spectrum01082-21 (Serological testing reveals the hidden COVID-19 burden among healthcare workers experiencing a SARS-CoV-2 nosocomial outbreak.)

Dear Dr. Yasutoshi Kido:

Thank you for submitting your manuscript to Microbiology Spectrum. When submitting the revised version of your paper, please provide (1) point-by-point responses to the issues raised by the reviewers as file type "Response to Reviewers," not in your cover letter, and (2) a PDF file that indicates the changes from the original submission (by highlighting or underlining the changes) as file type "Marked Up Manuscript - For Review Only". Please use this link to submit your revised manuscript - we strongly recommend that you submit your paper within the next 60 days or reach out to me. Detailed information on submitting your revised paper are below.

Link Not Available

Sincerely,

S. Wesley Long

Journals Department
Reviewer comments:

Reviewer #1 (Comments for the Author):

The purpose of this study is to assess the ability of SARS-CoV-2 antibody detection to supplement nucleic acid amplification testing (NAAT) for detection of COVID-19 infection in healthcare workers. In addition, the study aims to assess the relative risk of various aerosol-generating procedures by correlating apparent COVID-19 infection rates with self-reported workplace exposures. Serologic evidence of COVID-19 infection is also correlated with clinical symptoms or lack of symptoms.

The study has taken care to perform multiple SARS-CoV-2 antibody detection methods and to confirm positive results using a surrogate virus neutralization assay. Correlation of serologic data with symptom type (symptomatic vs asymptomatic vs hyposmia only) is a useful and relatively novel aspect of this study, as is the correlation of apparent COVID-19 infection rates with self-reported type of aerosol generated procedure exposure among healthcare workers. However, I have several significant concerns about the study design and major conclusions drawn.

Major comments:

- A primary conclusion of the study is that serologic testing identified an additional 27/64 total cases of COVID-19 infection that were missed by NAAT. I don't think this conclusion is supported by the data for several reasons:
 - o According to Table 1, NAAT results were not available for 4/27 individuals identified to have COVID-19 infection by serologic testing. We cannot conclude that NAAT failed to detect COVID-19 infection in these 4 cases, since it was never performed.
 - o Specimens collected for serologic testing lagged NAAT by 2 months. Given the possibility that subjects became infected with COVID-19 during the 2-month period between sample collections, serologic results cannot definitively comment on infection status at the time of NAAT collection/local outbreak. The authors state that the true spread of COVID-19 within the hospital was almost twice as extensive as that estimated using symptom-based NAAT. I don't think this statement is supported by the data given the possibility of COVID-19 infection acquired outside the hospital setting.
- Only 186/414 subjects underwent NAAT, while all subjects underwent serologic testing. To draw direct comparisons between the clinical sensitivities of NAAT and serology, all subjects must be tested using both techniques. Of the 350 cases with negative serologic results, 224 did not have NAAT results available. Isn't it possible that NAAT may have identified COVID-19 infection in some of the 224 cases? Similarly, serologic test results were not available for the 37 cases in which NAAT was positive (Table 1). It is possible that serologic results were negative in some of these 37 cases.
- In light of the above comments, I suggest the authors present a comparison of NAAT and serologic test results for the subset of cases in which both tests were performed.
- The authors suggest that symptom-based NAAT may miss COVID-19 infections that may be detected by serologic testing. I agree with this statement in principle but don't think the authors have sufficiently acknowledged the point that NAAT and serology detect fundamentally different things, that being current infection (NAAT) vs current or prior infection (serology). Due to the 2-month gap mentioned above, the authors have not provided any definitive data to suggest that a combination of serology and NAAT improves clinical sensitivity for COVID-19 infection over NAAT alone during the acute stage of infection.
- If serologic testing were used along with NAAT in a larger COVID-19 testing strategy within healthcare settings, as the authors suggest, how should healthcare workers with positive serology and negative NAAT be treated? Since this pattern of results would suggest prior infection, but not necessarily current infection, does serologic testing add anything more than epidemiological information?

Minor comments

- Nucleic acid amplification testing is more commonly abbreviated NAAT; I suggest updating NAT to

NAAT throughout the manuscript.

Reviewer #2 (Comments for the Author):

It is critical that you define the clinical sensitivity and specificity of the NAT and algorithm used in determination of cases. In addition, discussion should include that it is possible that not all cases that were identified serologically occurred during the outbreak, but that some might have been exposed before/after it and have therefore produced antibodies. Thereby making definite statements of the performance of NAT is not advisable either.

Staff Comments:

Preparing Revision Guidelines

For complete guidelines on revision requirements, please see the Instructions to Authors at [link to page]. **Submissions of a paper that does not conform to Microbiology Spectrum guidelines will delay acceptance of your manuscript.**

Please return the manuscript within 60 days; if you cannot complete the modification within this time period, please contact me. If you do not wish to modify the manuscript and prefer to submit it to another journal, please notify me of your decision immediately so that the manuscript may be formally withdrawn from consideration by Microbiology Spectrum.

If you would like to submit an image for consideration as the Featured Image for an issue, please contact Spectrum staff.

The purpose of this study is to assess the ability of SARS-CoV-2 antibody detection to supplement nucleic acid amplification testing (NAAT) for detection of COVID-19 infection in healthcare workers. In addition, the study aims to assess the relative risk of various aerosol-generating procedures by correlating apparent COVID-19 infection rates with self-reported workplace exposures. Serologic evidence of COVID-19 infection is also correlated with clinical symptoms or lack of symptoms.

The study has taken care to perform multiple SARS-CoV-2 antibody detection methods and to confirm positive results using a surrogate virus neutralization assay. Correlation of serologic data with symptom type (symptomatic vs asymptomatic vs hyposmia only) is a useful and relatively novel aspect of this study, as is the correlation of apparent COVID-19 infection rates with self-reported type of aerosol generated procedure exposure among healthcare workers. However, I have several significant concerns about the study design and major conclusions drawn.

Major comments:

- A primary conclusion of the study is that serologic testing identified an additional 27/64 total cases of COVID-19 infection that were missed by NAAT. I don't think this conclusion is supported by the data for several reasons:
 - o According to Table 1, NAAT results were not available for 4/27 individuals identified to have COVID-19 infection by serologic testing. We cannot conclude that NAAT failed to detect COVID-19 infection in these 4 cases, since it was never performed.
 - o Specimens collected for serologic testing lagged NAAT by 2 months. Given the possibility that subjects became infected with COVID-19 during the 2-month period between sample collections, serologic results cannot definitively comment on infection status at the time of NAAT collection/local outbreak. The authors state that the true spread of COVID-19 within the hospital was almost twice as extensive as that estimated using symptom-based NAAT. I don't think this statement is supported by the data given the possibility of COVID-19 infection acquired outside the hospital setting.
- Only 186/414 subjects underwent NAAT, while all subjects underwent serologic testing. To draw direct comparisons between the clinical sensitivities of NAAT and serology, all subjects must be tested using both techniques. Of the 350 cases with negative serologic results, 224 did not have NAAT results available. Isn't it possible that NAAT may have identified COVID-19 infection in some of the 224 cases? Similarly, serologic test results were not available for the 37 cases in which NAAT was positive (Table 1). It is possible that serologic results were negative in some of these 37 cases.
- In light of the above comments, I suggest the authors present a comparison of NAAT and serologic test results for the subset of cases in which both tests were performed.
- The authors suggest that symptom-based NAAT may miss COVID-19 infections that may be detected by serologic testing. I agree with this statement in principle but don't think the authors have sufficiently acknowledged the point that NAAT and serology detect fundamentally different things, that being current infection (NAAT) vs current or prior infection (serology). Due to the 2-month gap mentioned above, the authors have not provided any definitive data to suggest that a combination of serology and NAAT improves clinical sensitivity for COVID-19 infection over NAAT alone during the acute stage of infection.
- If serologic testing were used along with NAAT in a larger COVID-19 testing strategy within healthcare settings, as the authors suggest, how should healthcare workers with positive

serology and negative NAAT be treated? Since this pattern of results would suggest prior infection, but not necessarily current infection, does serologic testing add anything more than epidemiological information?

Minor comments

- Nucleic acid amplification testing is more commonly abbreviated NAAT; I suggest updating NAT to NAAT throughout the manuscript.

August 18, 2021

Dr. Yasutoshi Kido
Osaka City University
Osaka
Japan

Re: Spectrum01082-21R1 (Serological testing reveals the hidden COVID-19 burden among healthcare workers experiencing a SARS-CoV-2 nosocomial outbreak.)

Dear Dr. Yasutoshi Kido:

Your manuscript has been accepted, and I am forwarding it to the ASM Journals Department for publication. You will be notified when your proofs are ready to be viewed.

Sincerely,

S. Wesley Long
Editor, Microbiology Spectrum
